# TurboID-Based IRE1 Interactome Reveals Participants of the Endoplasmic Reticulum-Associated Protein Degradation Machinery in the Human Mast Cell Leukemia Cell Line HMC-1.2

**DOI:** 10.3390/cells13090747

**Published:** 2024-04-25

**Authors:** Nabil Ahmed, Christian Preisinger, Thomas Wilhelm, Michael Huber

**Affiliations:** 1Institute of Biochemistry and Molecular Immunology, Medical Faculty, RWTH Aachen University, 52074 Aachen, Germanytwilhelm@ukaachen.de (T.W.); 2Proteomics Facility, Interdisciplinary Centre for Clinical Research (IZKF), RWTH Aachen University, 52074 Aachen, Germany; cpreisinger@ukaachen.de

**Keywords:** ERAD pathway, mastocytosis, metadherin, ubiquitination, valosin-containing protein

## Abstract

The unfolded protein response is an intricate system of sensor proteins in the endoplasmic reticulum (ER) that recognizes misfolded proteins and transmits information via transcription factors to either regain proteostasis or, depending on the severity, to induce apoptosis. The main transmembrane sensor is IRE1α, which contains cytoplasmic kinase and RNase domains relevant for its activation and the mRNA splicing of the transcription factor XBP1. Mast cell leukemia (MCL) is a severe form of systemic mastocytosis. The inhibition of IRE1α in the MCL cell line HMC-1.2 has anti-proliferative and pro-apoptotic effects, motivating us to elucidate the IRE1α interactors/regulators in HMC-1.2 cells. Therefore, the TurboID proximity labeling technique combined with MS analysis was applied. Gene Ontology and pathway enrichment analyses revealed that the majority of the enriched proteins are involved in vesicle-mediated transport, protein stabilization, and ubiquitin-dependent ER-associated protein degradation pathways. In particular, the AAA ATPase VCP and the oncoprotein MTDH as IRE1α-interacting proteins caught our interest for further analyses. The pharmacological inhibition of VCP activity resulted in the increased stability of IRE1α and MTDH as well as the activation of IRE1α. The interaction of VCP with both IRE1α and MTDH was dependent on ubiquitination. Moreover, MTDH stability was reduced in IRE1α-knockout cells. Hence, pharmacological manipulation of IRE1α–MTDH–VCP complex(es) might enable the treatment of MCL.

## 1. Introduction

Mast cells (MCs) are crucial regulators of inflammatory processes via the production and release of various mediators (e.g., histamine, proteases, arachidonic acid metabolites, cytokines, chemokines, and growth factors [1]). Abnormal growth and/or activation of MCs in various tissues is pathophysiological. Systemic mastocytosis (SM) is a rare and heterogeneous disorder characterized by the unusual accumulation of MCs in the bone marrow or in multiple organ systems including the skin, digestive tract, and liver, amongst others [2,3]. Patients diagnosed with the indolent form of SM mainly suffer from the effects of hyper-release of MC mediators, while the aggressive form of SM is closely linked to the development of organ damage and a significantly diminished overall survival rate.

MC leukemia (MCL) is the most aggressive form of SM, and is characterized by a dramatic increase in MC numbers in the bone marrow and peripheral blood. The presence of KIT^D816V^ is a prominent characteristic mutation observed in over 80% of patients with SM, resulting in constitutive, ligand/stem cell factor-independent activation of the receptor tyrosine kinase KIT. In addition, KIT^D816V^ can be detected in other myeloid lineages in advanced stages of SM [4,5,6]. Although KIT^D816V^ is resistant to the tyrosine kinase inhibitor (TKI) imatinib, several TKIs (e.g., midostaurin) [7] have been identified as potent inhibitors. Nevertheless, therapy using TKIs is often accompanied by severe side effects and acquired resistance. Therefore, additional strategies to target neoplastic MCs have to be developed. 

In particular, in situations of enhanced proliferation and/or protein production, such as the growth of cancer cells and cytokine secretion from immune cells, adaptive control of proteostasis is essential. Previously, we have shown that the unfolded protein response (UPR) is a pro-survival pathway of the MCL cell line HMC-1.2 [8]. 

The UPR utilizes the ER stress sensors inositol-requiring enzyme 1α (IRE1α; gene name: *ERN1*), eukaryotic initiation factor 2α (eIF2α) kinase (PERK), and activating transcription factor 6 (ATF6) to recognize the accumulation of misfolded proteins in the ER. IRE1α is an ER-located type I transmembrane protein with catalytic endonuclease (RNase) and kinase activities in its cytosolic domain [9]. In response to ER stress, IRE1α cleaves the mRNA coding for the X-box-binding protein 1 (*XBP1*), resulting in its spliced form (*XBP1s*), which encodes an active transcription factor. XBP1s translocates into the nucleus and controls the transcription of a large number of UPR target genes, amongst them are genes encoding proteins involved in ER-associated protein degradation (ERAD) [10,11]. In the course of the ERAD process, misfolded luminal and transmembrane ER proteins are retro-translocated from the ER, which are then ubiquitylated proteins are extracted and then degraded within the 26S proteasome [12]. The molecular force required for extraction and unfolding is provided by valosin-containing protein (VCP), which belongs to the group of AAA ATPases, and can be seen as the main executor of the ERAD machinery [13]. The ER transmembrane protein metadherin (MTDH) has been characterized as an oncogene that affects several signaling pathways and transcription factors, such as RAS, MYC, PI3K/AKT, NFκB, mitogen-activated protein kinase, and WNT [14]. It has been identified as a functional driver in multiple aspects of cancer progression, promoting cancer cell proliferation and resistance to treatment by acting as an RNA-binding protein [15]. Most likely, misfolded MTDH or homeostatically removed MTDH is processed by the activity of VCP, suggesting functional and/or local interactions between these proteins.

Depending on the strength and/or duration of the ER stress, IRE1α is activated by dimerization or oligomerization. Moreover, several interactors have been identified to regulate the function of IRE1α [16]. Hence, IRE1α in a context-driven manner can regulate the adaptive arm of the UPR as well as promote the terminal UPR leading to apoptosis [9]. In contrast to the ubiquitously expressed IRE1α, its paralogue, IRE1β (gene name: *ERN2*), is considered to be specifically expressed in the epithelial cells of mucosal surfaces and is a less efficient transducer of the UPR, and may function as a negative regulator of IRE1α [17,18].

However, despite the substantial contribution of IRE1α in maintaining ER homeostasis and the knowledge on its numerous interactors, there remains a considerable gap in the comprehension of its functions in the cancer framework, necessitating further investigation. Recently, several studies reported that IRE1α is an essential regulator of migration, invasion, and metastasis in diverse types of cancer [19,20,21]. Consequently, further investigation is needed to enhance our understanding of the biological role of IRE1α in cancer progression and its associated pathological mechanisms [22].

In this study, we aimed to identify IRE1α and IRE1β interaction partners that are involved in regulation of IRE1 function in the MCL model system of HMC-1.2 cells. We conducted an independent proteomic screen utilizing proximity-dependent biotin identification (TurboID) using IRE1 as the bait. Our analysis showed that the predominant interacting proteins in HMC-1.2 cells were consistently bound to IRE1α and mostly unaffected by the presence of additional ER stress conditions. In particular, we found that the stability of IRE1α and the ER transmembrane protein MTDH is dependent on VCP. Furthermore, the presence of IRE1α could strengthen the initial stability of MTDH which might play a crucial role in the progression of MCL.

## 2. Materials and Methods

### 2.1. Cell Culture

HMC-1.2 (KIT^V560G,D816V^) and HMC-1.1 (KIT^V560G^) cells were kindly provided by Dr. J. Butterfield (Mayo Clinic, Rochester, MN, USA) [23]. T84 cells used in this study were generously provided by Prof. Dr. Jörg Schulzke (Charité, Berlin, Germany). Jurkat, K562 and HEK-293T were acquired from ATCC (American Type Culture Collection, Manassas, VA, USA). 

T84 cells were grown in DMEM-F12 (Gibco, Thermo Fisher Scientific, Waltham, MA, USA, #10565018) and HEK293 in DMEM (Gibco, Thermo Fisher Scientific, Waltham, MA, USA, #10569010) supplemented with 10% fetal bovine serum (FBS) (Capricorn, Duesseldorf, Germany #FBS-12A), 50 units/mL penicillin and 50 ng/mL streptomycin (Sigma-Aldrich, Darmstadt, Germany, #P4333). All other cell lines were cultured in RPMI 1640 medium (Invitrogen, Dreieich, Germany, #21875-0991) supplemented with 10% FBS, 50 units/mL penicillin and 50 ng/mL streptomycin in an atmosphere containing 5% CO_2_. The medium was renewed twice a week. 

### 2.2. Plasmid Constructs

General molecular biology reagents, including restriction enzymes, Phusion HF DNA Polymerase (#, M0530) and Quick Ligation Kit (#M2200), were from New England Biolabs (Frankfurt, Germany). DH5α competent cells were provided by the Institute of Biochemistry, Hospital University of Aachen. QIAquick gel extraction (#M0530) and DNA purification kits (#51306/#12662) were from Qiagen (Hilden, Germany). 

The following plasmid were used in this study:

Flag_HsIRE1a_pBabePuro was a gift from David Ron (Addgene plasmid #54337; http://n2t.net/addgene:54337 (accessed on 22 February 2019); RRID:Addgene_54337) and has been described in Volmer et al. [24]. Flag_HsIRE1α_delta589-903 was generated from Flag_HsIRE1α_pBabePuro. The IRE1α kinase-dead (KD) construct hIRE1α K599A was a gift from Fumihiko Urano (Addgene plasmid #20745; http://n2t.net/addgene:20745 (accessed on 22 February 2019); RRID: Addgene_20745) [25].

pBabe-puro was a gift from Hartmut Land and Jay Morgenstern and Bob Weinberg (Addgene plasmid #1764; http://n2t.net/addgene:1764 (accessed on 22 February 2019); RRID: Addgene_1764) [26].

V5-TurboID-NES_pCDNA3 was a gift from Alice Ting (Addgene plasmid #107169; http://n2t.net/addgene:107169 (accessed on 2 April 2024); RRID: Addgene_107169) [27].

pUMVC (Addgene plasmid # 8449; http://n2t.net/addgene:8449 (accessed on 24 April 2020); RRID: Addgene_8449) and pCMV-VSV-G (Addgene plasmid #8454; http://n2t.net/addgene:8454 (accessed on 2 April 2024); RRID: Addgene_8454) were a gift from Bob Weinberg [28]

pSpCas9(BB)-2A-Puro (PX459) V2.0 was a gift from Feng Zhang (Addgene plasmid #62988; http://n2t.net/addgene:62988 (accessed on 2 April 2020); RRID: Addgene_62988) [29].

VCP(wt)-EGFP was a gift from Nico Dantuma (Addgene plasmid #23971; http://n2t.net/addgene:23971 (accessed on 2 April 2024); RRID: Addgene_23971) [30].

Human *ERN2* construct coding sequences were PCR amplified from *ERN2* templates (provided by K. Kohno, Nara Institute of Science and Technology, Ikoma, Japan) [31] using primers with NaeI and AgeI restriction sites for the insertion into pBabe-puro [26].

Expression constructs coding for *ERN1* or *ERN2* fused to C-terminal TurboID were generated by restriction enzyme-based molecular cloning using the AgeI and XhoI restriction sites from V5-TurboID-NES_pCDNA3 [27]. 

CRISPR–Cas9-mediated gene knockouts of human *ERN1* and *ERN2* were performed using the pSpCas9(BB)-2A-Puro (PX459) V2.0 plasmid (Ran et al., 2013) containing gRNA for IRE1α (5′-CTTTTATGTCTGGCAGCGGG-3′) and gRNA for IRE1β (5′-GCTGAGCAACTCACCGTAGT-3′). Pairs of oligonucleotides with a BbsI overhang were annealed and then ligated into the BbsI-digested vector.

### 2.3. Mammalian Cell Transfections

Transient plasmid transfections of HEK293 cells were performed using the transfection reagent TransIT-LT1 (MirusBio, Madison, WI, USA, #731-0029) according to the manufacturer’s instructions. Generally, 1 µg of MSCV-GFP plasmid encoding GFP was co-transfected as a control to verify that the transfection efficiency was around 80% in all wells. Transfected cells were usually cultured for 24–48 h before further manipulation or analysis. The stable cell lines were generated by retroviral transductions. First, HEK293T cells were transfected with a mixture of DNA containing 1 μg of pBABE-puro target, 0.9 μg of pUMVC packaging, and 0.1 μg pCMV-VSV-G envelope plasmids using the TransIT-LT1 transfection reagent. The supernatant of the transfected cells was collected 48 h later and filtered through a 0.45 μm pore filter. For viral infection, the virus-containing supernatants from HEK293T cultures were mixed with polybrene (Sigma-Aldrich, Darmstadt, Germany, #TR-1003) to a final concentration of 8 µg/mL and were added to HMC-1.2 or HEK293 cells. Together with the viral supernatant, the cells were centrifuged at 1000 g for 1.5 h at room temperature (RT) and the medium was renewed. This step was repeated two more times every 24 h. Then, the cells were returned to an CO_2_ incubator, and 72 h later, the cells were selected in 2.5 µg/mL puromycin (Sigma-Aldrich, #P8833).

The cell lines HMC-1.2^Flag-*hIRE1α*^, HMC-1.2*^hIRE1α^*^-TurboID^, HMC-1.2*^hIRE1β^*^-TurboID^, and HMC-1.2^V5-TurboID^ were generated by stable transduction of HMC-1.2*^IRE1^*^−/−^ cells. HEK293^Flag-IRE1α^ was generated by stable transduction of HEK293*^IRE1α^*^−/−^ cells.

### 2.4. Pharmacological Inhibitors and ER Stress Inducers

To induce ER stress, cells were exposed to 5 µg/mL tunicamycin (TM, Applichem, Darmstadt, Germany, #A2242), 100 nM thapsigargin (Tg, Abcam, Cambridge, UK, #ab120286), or 10 nM bortezomib (BZ, Selleckchem, Cologne, Germany, #S1013) for the indicated period. 20 µM of the E1 ubiquitin-activating enzyme inhibitor MLN7243 (MedChemExpress, Junction, NJ, USA, #HY-100487) was used to prevent ubiquitination.

For the cycloheximide (CHX) chase experiments, HMC-1.2 cells were treated with 50 μg/mL CHX (Sigma-Aldrich, #01810) for the indicated times. VCP activity was inhibited by CB-5083 (#19311) or NMS-873 (#17674) (1 μM; both from Cayman, Ann Arbor, MI, USA).

### 2.5. RNA Preparation and Quantitative RT-PCR

RNA from 2 × 10^6^ HMC-1.2 cells was extracted using a NucleoSpin RNA Plus Kit (Macherey Nagel, Dueren, Germany, #740955.50) according to the manufacturer’s instructions. Total RNA (1 μg) was reverse transcribed using random oligonucleotides (Roche, Merck, Darmstadt, Germany #11034731001) and an Omniscript Kit (Qiagen, #205113) according to the manufacturer’s instructions. qPCR was performed on a Rotorgene (Qiagen) by using the SYBR green reaction mix SensiFast (Meridian Biosciences, Cincinnati, OH, USA, #QT650-02). The primers used for PCR amplification were as follows: 

*XBP1s* fwd CTGAGTCCGAATCAGGTGCAG, rev: GAAGAGTCAATACCGCCAGAAT; *DNAJB4* fwd GGAAGGAGGAGCGCTAGGTC, rev ATCCTGCACCCTCCGACTAC; *HSPA5* fwd TGACATTGAAGACTTCAAAGCT, rev CTGCTGTATCCTCTTCACCAGT; *ATF4* fwd GCTAAGGCGGGCTCCTCCGA, rev ACCCAACAGGGCATCCAAGTCAA; *CHOP* fwd GGAGCATCAGTCCCCCACTT, rev TGTGGGATTGAGGGTCACATC; *PPP1R15A* fwd GAGGAGGCTGAAGACAGTGG, rev AATTGACTTCCCTGCCCTCT; *ATF6* fwd CGAAGGGATCACCTGCTGTT rev, CCTGGTGTCCATCACCTGAC; *XBP1u* fwd CAGCACTCAGACTACGTGCA, rev ATCCATGGGGAGATGTTCTGG; *EDEM1* fwd CCACTGAGACCAGACTTAGTG, rev CGTACCCACACTTGACTTTTGTG; *ERN1* fwd TGCTTAAGGACATGGCTACCATCA rev, CTGGAACTGCTGGTGCTGGA; *ERN2* fwd TCGAAGGACCAATGTACGTCA, rev GGATGGTGAATGGCAGTTTCAT; *HPRT* fwd TGACACTGGCAAAACAATGCA, rev GGTCCTTTTCACCAGCAAGCT. All quantitative RT-PCR reactions were performed in triplicate. Transcript expression was normalized to the housekeeping gene *HPRT* and relative expression was calculated according to the delta-CT method [32]. 

### 2.6. TurboID, Biotinylation Assay, and Mass Spectrometry

TurboID was performed as described previously [27] with minor modifications. In the beginning, 30 × 10^6^ HMC-1.2^IRE1α−/−^ cells expressing different constructs of IRE1α-TurboID fusion proteins were cultivated in medium supplemented with the solvent control DMSO or tunicamycin for 2 h. Subsequently, 500 µM biotin (Sigma-Aldrich, #B4501) was added to the medium to promote proximity biotinylation for 1 h. After the incubation period, the cells were rinsed three times in phosphate-buffered saline (PBS) and then subjected to lysis using RIPA buffer (50 mM Tris-HCl pH 7.5, 250 mM NaCl, 1% Triton X-100, 1 mM DTT, 0.5% Na-deoxycholate, and 1mM PMSF), supplemented with protease inhibitors (20 µg/mL aprotinin, 4 µg/mL leupeptin) and incubated on a rotator for 1 h at 4 °C. The lysates were then transferred into 1.5 mL reaction tubes and sonicated in an ice bath three times with a 30% duty cycle for 20 s with 1.5 min breaks on ice. The suspension was centrifuged for 15 min at 4 °C and 16.500× *g* to remove the cell debris and the clear supernatant was applied to a PD-10 desalting column (Cytiva, Dreieich, Germany, #17085101) to remove the excess free biotin using the gravity protocol according to the manufacturer’s instructions. Proteins were eluted with 1 mL of lysis buffer. The protein concentration of the protein extract was measured using a BCA protein assay kit (Pierce™, Thermo Fisher Scientific, #23225). To enrich for biotinylated proteins from the protein extracts, 250 µL of streptavidin magnetic beads (Pierce™, Thermo Fisher Scientific, #88817) were washed twice with wash buffer (lysis buffer without protease inhibitors and PMSF), and desalted lysates containing 3 mg protein were then incubated with the beads on a rotator overnight at 4 °C. The beads were washed twice with 1 mL of wash buffer. To completely remove the detergent, the beads were washed once in 1 mL of 1 M KCL for 2 min, then 0.1 M Na_2_CO_3_ for 10 s, and lastly with 2 M urea in 10 mM Tris-HCL pH 8.0 for 10 s. Finally, the beads were resuspended in 1 mL of final buffer (50 mM Tris-HCl pH 7.5 and 250 mM NaCl). To confirm the successful enrichment of the biotinylated proteins, 5% of the washed streptavidin beads were used for analyses by silver staining gel, and the remaining beads were flash-frozen in liquid nitrogen and stored at −80 °C until the LC-MS/MS analysis.

For the mass spectrometry analysis, the co-immunoprecipitates/TurboID precipitates were washed three times with lysis buffer w/o detergent. The samples were then prepared as described previously [33]. The resulting lyophilized peptides were resuspended in 15 µL of 3% formic acid (FA)/5% acetonitrile (ACN) and 5 µL were loaded onto a nanoLC system (RSLCnano, Thermo Fisher Scientific, Bremen, Germany). Prior to separation, the peptides were subjected to trapping on a pre-column (Acclaim PepMap100, C18, 5 µm, 100 Å, 300 µm i.d. × 5 mm, Thermo Fisher Scientific, Bremen, Germany). The analytes were then separated on an analytical column (Easyspray 50 cm column (ES803) at 45 °C; Thermo Fisher Scientific, Bremen, Germany) using a 70 min gradient run at 250 nL/min (0–10 min: 5% buffer B (buffer A: 0.1% FA; buffer B: 80% acetonitrile, 0.1% FA); 10–42 min: 5–35% buffer B; 42–47 min: 35–99% buffer B; 47–52 min: 99% buffer B; 52–55 min: 99–5% buffer B; 55–70 min: 5% buffer B). The eluting peptides were analyzed on a Q Exactive plus mass spectrometer (250 °C capillary temperature, 2 kV spray voltage; Thermo Fisher Scientific, Bremen, Germany) in data-dependent mode. The full MS settings were as follows: resolution, 70,000; AGC target, 2 × 105; maximum injection time, 100 milliseconds; scan range, 300–1650 *m*/*z*. The dd-MS2 settings were as follows: resolution, 17,500; AGC target, 2 × 105; maximum injection time, 110 milliseconds; top 10 precursor fragmentation; isolation window, 1.8 *m*/*z*; collision energy, 27. The dd settings were as follows: minimum AGC, 5 × 102; 10 s dynamic exclusion; only 2+ to 5+ peptides were allowed.

The raw data from the Orbitrap instrument were analyzed with MaxQuant (MQ, version 2.0.1.0) with the built-in Andromeda search engine [34]. The search was conducted against the human UniProt database (version 04/2021; with only reviewed and canonical sequences being used) and MQ default settings. The specific settings were protease: trypsin (two missed cleavages allowed); fixed modification: carbamidomethylation (Cys); variable modifications: oxidation (Met) and N-terminal protein acetylation; false discovery rate: 0.01 (both peptide and protein levels); minimum peptide length: seven amino acids. Relative quantification was carried out by utilizing the label-free quantitation algorithm from MQ.

The resulting proteinGroups.txt file was further analyzed using the Perseus suite (version 1.6.14.0; [35]). The three biological replicates of the six experimental conditions (IRE1α, IRE1β, and con (plus DMSO or tunicamycin) were grouped into three groups (IRE1α, IRE1β, and con) resulting in six “replicates” per condition (see main text for explanation)). The protein list was then filtered for reversed hits, contaminants, and “only identified by site” entries. Furthermore, a protein was required to be identified by a minimum of two unique peptides for further evaluation. The data were log2-transformed and filtered for proteins that were identified in all six replicates of one of the three groups; afterwards, standard imputation of missing values was performed. An additional analysis was carried out using the two-sample tests feature in Perseus. These files were then used to generate the volcano plot (Instant Clue, [36], Venn diagram (https://bioinfogp.cnb.csic.es/tools/venny/ (accessed on 2 April 2024)) and string enrichment analysis (Cytoscape 3.10.0; stringApp 2.0.1).

### 2.7. Immunoprecipitation, Immunoblotting, and Antibodies

Cells were lysed in RIPA buffer and centrifuged for 10 min at 16.500× *g*. The supernatant of the lysates were then incubated with anti-Flag M2 Magnetic beads (Sigma, #M8823) or anti-GFP-Trap Magnetic Agarose (ChromoTek, Proteintech, Planegg-Martinsried, Germany, #gtma) for the indicated period at 4 °C. Subsequently, the beads were washed five times in wash buffer and eluted by boiling in 2× SDS sample buffer. The protein samples were separated by SDS-PAGE and transferred to PVDF membranes (Thermo Fisher Scientific, #88518). The blots were blocked for 30 min at RT in TBS-T buffer (50 mM Tris (pH 7.4), 150 mM NaCl, 0.1% Tween-20) containing 5% bovine serum albumin and then incubated with the selected antibody solutions. Immunoblot analysis was performed with the indicated antibodies and visualized by chemiluminescence (ImageQuant™ LAS 4000, GE Healthcare, Chicago, IL, USA). The following antibodies were used: anti-IRE1α (#14C10), anti-BIP (#3183), anti-VCP (#2648), and anti-ubiquitin (#3936), which were purchased from Cell Signaling Technology (Leiden, The Netherland); anti-MTDH (#40-6500), anti-VCP (#MA3-004), and anti-streptavidin HRP (#21130), which were purchased from Thermo Fisher Scientific; anti-phospho-IRE1α S724, which was purchased from Novus (#2323SS); anti-GAPDH (#sc-32233), which was from Santa Cruz Biotechnology (Heidelberg, Germany); anti-vinculin (#V9131) and anti-FLAG M2 (#F1804), which were from Sigma-Aldrich (Darmstadt, Germany); anti-GFP, which was from Rockland (#600-101-215); anti-VAPA which was from Ptglab (ChromoTek, Martinsried, Germany, 15275-1-AP); and anti-emerin, which was a gift from Wolfram Antonin (Institute of Biochemistry and Molecular Cell Biology). HRP-coupled secondary antibodies were purchased from Agilent (Dako, Waldbron, Germany, #P0448 and P0447). 

### 2.8. Densitometric Analysis

To quantitatively analyze the protein bands from the Western blot images, ImageJ software 1.53k was used following a structured procedure. The recommendations of the National Institutes of Health (NIH) protocol (https://imagej.net/ij/nih-image/manual/tech.html (accessed on 2 April 2024)) were followed. The obtained quantitative data were exported to the Excel program for further analysis. Normalization was achieved by dividing the intensities of the test protein (e.g., MTDH) by the respective intensities of the control protein (e.g., vinculin). This process assists in normalizing the data across samples and eliminates any background signals from the protein signal. Comparative expression levels were determined by calculating the ratio of normalized values between test samples (after treatment) and control sample (untreated), which were then standardized to a value of 1 (untreated sample).

### 2.9. Immunofluorescence (IF)

HEK-293 cells were cultured in twelve-well plates with glass coverslips until they reached 30–40% confluency. The cells were washed and fixed with 4% paraformaldehyde, permeabilized with 0.1% Triton X-100 in PBS, and blocked with 5% bovine serum albumin in PBS containing 0.1% Triton X-100. Then, the cells were incubated with rabbit anti-calnexin (#2679, CST) primary antibody at 4 °C overnight. After washing, the cells were stained with anti-DYKDDDDK tag (#MA1-142-A488, Invitrogen) to detect Flag-IRE1α and Alexa Fluor 594-conjugated anti-rabbit IgG (A-11012, Invitrogen) to detect calnexin for 1 h at room temperature. After washing, fluorescent signals were detected under a confocal microscope [FV1000; Olympus, Hamburg, Germany; equipped with a photomultiplier (model R7862; Hamamatsu)]. 

### 2.10. Statistical and Bioinformatic Analysis

All data shown were generated from at least three independent experiments. The statistical analysis and graphing of data were performed using GraphPad Prism 10 Version 10.2.2 (GraphPad Software, San Diego, CA, USA, 92108). All statistical test procedures were performed as described in the respective figure legends. *p*-values were considered statistically significant according to the GP style in GraphPad Prism (ns: *p* > 0.05, * *p* < 0.05, ** *p* < 0.01, *** *p* < 0.001, **** *p* < 0.0001). The respective number of independent biological replicates per experiment are indicated in the figure legends. To address the quantitative aspects of the proteomic data, we utilized the MetaboAnalystR package (version 3.1.0) to compare the label-free quantitation (LFQ) intensity between the IRE1 groups and the control group. In addition to the comparative analysis, we employed the UniProtR package (version 2.2.2), which allowed us to perform a comprehensive Gene Ontology (GO) analysis, enriching our understanding of the biological pathways affected by the observed perturbations.

## 3. Results

### 3.1. UPR Activation in IRE1α- and IRE1β-Deficient HMC-1.2 Cell Lines

Maintaining proteostasis is particularly essential for the proliferation and survival of cancer cells, which are therefore strongly dependent on a functional UPR. We have shown previously that the induction of proteotoxic ER stress in the MCL cell line HMC-1.2 activates the mechanisms of a terminal UPR [8]. Although the RNase domain of IRE1α is the main executer of *XBP1* splicing, a significant amount of spliced *XBP1* was still detectable in IRE1α-deficient cells upon induction of ER stress using thapsigargin (Tg) (Appendix A). The Tg-induced production of PERK-dependent *CHOP* (*DDIT3*) was not affected by IRE1α deficiency (Appendix A). This might suggest a selective role of the second IRE1 isoform, IRE1β. To date, IRE1β expression was only reported in non-ciliated cells, such as intestinal epithelial cells and airway mucous cells in humans [17]. Thus, we analyzed the relative mRNA expression of IRE1β in different human cell lines, including human MC lines. While IRE1β was notably detectable in the human colon carcinoma cell line T84 [18], a lower amount of mRNA expression was detectable in all analyzed cell lines, suggesting a potential role in HMC-1.2 cells (Appendix A).

To elucidate the biological implications of both IRE1 isoforms in an MCL model system, single IRE1α-, single IRE1β-, and double IRE1α/β-deficient HMC-1.2 cells were generated using CRISPR/Cas9 gene editing. Importantly, compared to the parental cells, the expression of IRE1β mRNA was strongly reduced in both deficient cell lines, HMC-1.2^IRE1β−/−^ and HMC-1.2^IRE1α/β−/−^ (Appendix A), proving the specificity of the primers used for the RT-qPCR analysis.

Tunicamycin (TM), a potent ER stressor that inhibits N-linked glycosylation and thus causes aggregation of misfolded membrane or secreted proteins, was used to evaluate the ER stress response in the IRE1-deficient cell variants compared to the parental HMC-1.2 cells. The expression of UPR target genes measured by RT-qPCR confirmed that TM was able to activate all three arms of the UPR (IRE1α, PERK, and ATF6) in the parental HMC-1.2 cells. TM activated the target genes in the IRE1 pathway (*XBP1s*, *DNAJB4*, and *HSPA5*), PERK pathway (*ATF4*, *CHOP*, and *PPP1R15A*), and ATF6 pathway (*ATF6*, *us-XBP1*, and *EDEM1*). Notably, *HSPA5* is a target gene of the ATF6 pathway as well. IRE1-dependent splicing of *XBP1* (Figure 1A) and the expression of the XBP1s target gene *DNAJB4* (Figure 1B) were strongly reduced in IRE1α single- as well as IRE1α/β double-deficient cells. Interestingly, the single deletion of IRE1β alone also reduced *XBP1* splicing, suggesting a contribution of IRE1β in *XBP1* splicing while *DNAJB4* expression was not significantly affected. The expression of *HSPA5* (coding for BIP, Figure 1C), a common target gene of both XBP1s and ATF6f, was not affected by IRE1 depletion, suggesting a dominant role of ATF6f. Although IRE1 was depleted in the generated HMC-1.2 cell lines, the activation of the PERK and ATF6 pathways was not further enhanced by compensatory mechanisms, as measured by the expression of *ATF4* (Figure 1D), *DDIT3* (coding for CHOP, Figure 1E), and *PPP1R15A* (coding for GADD34, Figure 1F). While the increases in *ATF4* and *CHOP* expression induced by TM were comparable in the parental and IRE1-deficient cell lines, *PPP1R15A* was induced to a smaller degree in all IRE1-deficient cells, suggesting another function of IRE1α and/or IRE1β in MCL cells. Although *ATF6* (Figure 1G) expression was not increased after 3 h of TM treatment, the ATF6f target gene *us-XBP1* (Figure 1H) was significantly increased in all studied cell lines. Finally, the expression of the ERAD marker *EDEM1* was not induced by TM application in the cell lines under study (Figure 1I). In conclusion, IRE1-deficient HMC-1.2 cell lines showed a comparable activation of IRE1-independent UPR target genes. Interestingly, IRE1β appears to contribute to TM-induced *XBP1* splicing.

### 3.2. Identification of IRE1 Interacting Proteins Using a TurboID Approach

The accumulation of misfolded proteins in the ER causes the dimerization or oligomerization of IRE1, leading to the activation of its cytoplasmic kinase and RNase domains. Although this activation is mainly mediated by BIP dissociation, several interacting proteins have been identified in the regulation of IRE1 function [37]. 

The current study was designed to identify potential regulators of IRE1 function by employing the TurboID proximity labeling technique to discern IRE1α and IRE1β interaction proteomes in the HMC-1.2 cell line. Therefore, IRE1α-deficient HMC-1.2 (HMC-1.2^IRE1α−/−^) cells were stably transduced with IRE1 constructs containing the BirA-variant TurboID fused to the IRE1 C-terminus (Figure 2A). Initially, the subcellular location of overexpressed IRE1α-TurboID was validated in the ER using fluorescence microscopy in HEK293 cells, demonstrating co-staining of FLAG-tagged hIRE1α-TurboID with the ER marker calnexin (Appendix A). Moreover, ER stress-induced S724 phosphorylation of FLAG-tagged IRE1α-TurboID was comparable to the signal of heterologously expressed FLAG-tagged IRE1α (Appendix A). The expressed IRE1α-TurboID or IRE1β-TurboID fusion proteins were used as bait proteins to specifically focus on proteins that bind in the cytosol. The interactomes were compared to the cytosolic empty vector control V5-TurboID under non-stressed and ER stress (induced with TM for 2 h) conditions before biotin was supplemented for 1 h to facilitate proximal protein biotinylation (see Figure 2A for work pipeline). Biotinylated proteins were enriched using a streptavidin column. A small portion of enriched proteins was used to compare the respective patterns by silver staining before the samples were subjected to LC-MS/MS (Figure 2A).

The MS analysis identified a total number of 1232 proteins present in at least one sample. Following data processing, which included contaminant filtering and log2 transformation, we derived a final count of 624 proteins. The Label-Free Quantification (LFQ) intensity was determined for all identified proteins. The LFQ intensities showed a high Pearson correlation among replicates. However, the correlation coefficient between the IRE1α and IRE1β samples was approximately 0.5, reflecting their high similarity (Appendix A, Datafile S1). A Principal Component Analysis (PCA) of significant proteins indicated a distinct separation between the two IRE1 isoforms and the control group, albeit without a noticeable variance between unstressed and ER stress conditions (Figure 2B). Therefore, we decided to merge the DMSO- and TM-treated groups. Following an overlap analysis of the protein candidates identified in the TurboID/MS analysis, 194 IRE1-interacting proteins were identified within all IRE1 constructs under non- and ER-stressed conditions (Figure 2C). 

In addition, hierarchical clustering of the top 250 overrepresented proteins showed distinguishable patterns among the groups, especially between the cytoplasmic control and the IRE1 isoforms. Again, no clear separation between the DMSO and TM treatments were observed (Figure 2D), which supports our previous observation of an already activated UPR under basal conditions in HMC-1.2 cells [8] 

The major changes that met the criteria of the quantitative MS analysis between IRE1α and control (Figure 2E) and between IRE1ß and control (Appendix A) are depicted in volcano plots. 

Interestingly, the subsequent STRING analysis of the specific IREα interactors and Gene Ontology (GO) and pathway enrichment analyses of the candidate proteins revealed that the majority of the enriched proteins have an ER localization and are involved in ER stress responses and ER to Golgi vesicle-mediated transport (Figure 2D–F).

### 3.3. IRE1α Interacts with Proteins Involved in Vesicle Transport and Protein Stabilization

On the basis of the aforementioned GO terms (Appendix A), our next focus was directed toward three main classifications: ER to Golgi vesicle-mediated transport, negative regulators of apoptosis (oncoproteins), and the ERAD pathway. We generated box plot graphs for each protein of interest, focusing first on proteins in contact with the ER, such as VAPA, or the nuclear membrane, such as emerin. Additionally, we focused on the classified oncoproteins and RISC complex members MTDH and SND1. While VAPA (Figure 3A) and emerin (Figure 3B) were enriched in both IRE1α- and IRE1β-containing protein complexes, MTDH (Figure 3C), SND1 (Figure 3D), and the AAA ATPase VCP (Figure 3E) were only enriched in IRE1α-positive cells, suggesting differential regulation by interacting proteins of IRE1α and IRE1β. In contrast, the heat shock protein HSP90 showed a stronger interaction with IRE1β (Figure 3F).

These interactions were confirmed through co-immunoprecipitation experiments using FLAG-hIRE1α or an empty vector control (MOCK), both stably transduced in HMC-1.2^IRE1α−/−^ cells, followed by treatments with the ER stressor TM or the solvent control DMSO. The immunoblot analyses confirmed interactions between FLAG-tagged IRE1α and endogenous VAPA, Emerin, and MTDH but not with the MOCK control (Figure 3G). In order to validate the interaction between IRE1α and VCP, we performed co-immunoprecipitation assays with FLAG-tagged hIRE1α or GFP-tagged hVCP in HEK 293T cells. Consistently, we confirmed that VCP could be precipitated via GFP-specific antibodies (Figure 3H) and IRE1α could be precipitated via FLAG-specific antibodies (Figure 3I). These results not only validated the TurboID analyses, but also revealed the diversity in the protein interactions of IRE1α.

### 3.4. VCP Inhibition Stimulates UPR Activation and Enhancement of IRE1α Expression

Based on the observed interaction of VCP with IRE1α, we next investigated the consequences of VCP inhibition on IRE1α expression and UPR activation. A previous study demonstrated that VCP inhibition causes the activation of the UPR [38]. Two VCP inhibitors, CB-5803 (Figure 4A) and NMS-873 (Appendix A), were used to block the ATPase activity of VCP in HMC-1.2 cells. A significantly increased amount of IRE1α and BIP was detectable after 18 h of VCP inhibition (Figure 4A,B), which also resulted in the detection of increased phosphorylation of IRE1α at S724 (Figure 4A and Appendix A), suggesting the induction of ER stress by blockade of the ERAD mechanism. Additionally, *ERN1* (Figure 4C) and *XBPs* (Figure 4D) mRNA expression levels were increased by VCP inhibition together with *HSPA5* (Figure 4E) and *CHOP* mRNA (Figure 4F). The increased protein and mRNA levels suggest a cooperation between stress-mediated induction of transcription (and translation) as well as inhibitor-mediated blockade of homeostatic degradation. 

Interestingly, a similar activation of the UPR by VCP inhibition was observed in additional cancer cell lines, such as Jurkat and K562 cells, indicating a conserved mechanism across various cancer types (Appendix A). Collectively, these observations suggest that the inhibition of VCP in the MCL cell line HMC-1.2 induces ER stress and activates the UPR, particularly the IRE1α–XBP1s pathway, resulting in the increased expression of UPR target genes.

### 3.5. The Interaction between IRE1α, VCP, and MTDH Is Dependent on Ubiquitination

During ERAD, VCP associates with ubiquitin ligases located at the ER and extracts misfolded proteins from the ER [39]. To comprehensively understand the observed effects of VCP inhibition and the physical interaction between IRE1α and VCP, our investigation incorporated the interaction of IRE1α with MTDH, which is recognized as a type II ER transmembrane protein [40]. First, to analyze whether the interaction of IRE1α with VCP or MTDH is dependent on ER stress, we treated HMC-1.2^FLAG-IRE1α^ cells with the pharmacological ER stress inducers TM or bortezomib (BZ). Second, to study if the interaction of IRE1α with VCP or MTDH is dependent on VCP activity or ubiquitination, we treated the cells for 3 h with the VCP inhibitor CB-5803 or the inhibitor of the E1 ubiquitin-activating enzyme MLN7243. Subsequently, we conducted co-immunoprecipitation assays using anti-FLAG antibody-coupled beads (to precipitate FLAG-hIRE1α) in order to determine the effects of these treatments on the IRE1α–VCP and IRE1α–MTDH interactions (Figure 5A–C). Interestingly, the results of our study indicated that the interaction between IRE1α and VCP or IRE1α and MTDH is mostly dependent on ubiquitination, as demonstrated by the significantly reduced interaction in cells exposed to the E1 inhibitor MLN7243 (Figure 5A–C). The impact of MLN7243 on global protein ubiquitination in the parental HMC-1.2 cells is shown in Appendix A. In addition, we confirmed the ubiquitin-dependent interaction of endogenous IRE1α with VCP in the parental HMC-1.2 cells (Appendix A). Moreover, the IRE1α variant IRE1α_delta589-903 containing only Lys545 still interacted with VCP (Appendix A). In addition, the kinase-inactive variant IRE1α KD (K599A) also interacted with hVCP-GFP, excluding a role of the kinase activity for this interaction (Appendix A).

### 3.6. VCP Inhibition Stabilizes Expression of IRE1α and MTDH

As shown above, VCP inhibition by CB-5803 resulted in the increased expression of IRE1α and induction of ER stress (Figure 4A). We next analyzed the effect of pharmacological VCP inhibition on IRE1α and MTDH protein stability. Therefore cycloheximide (CHX) chase assays were performed, wherein the HMC-1.2 cells were incubated with the translation inhibitor CHX in the presence or absence of the VCP inhibitors, CB-5083 or NMS-873. The subsequent immunoblot analyses showed a gradual decrease in IRE1α expression over a 24 h period in DMSO-treated cells. In contrast to the DMSO-treated cells, in the presence of the VCP inhibitors, the half-life of IRE1α was significantly prolonged (Figure 5D,E and Appendix A). Compared to IRE1α, MTDH showed a distinct expression profile, with a rapid decrease already observed after 4 h. Similar to IRE1α, the half-life of MTDH was significantly prolonged by the treatment with CB-5083 (Figure 5D,F and Appendix A). These results suggest that both IRE1α [41] and MTDH are substrates for ERAD, with their stability directly affected by VCP activity. 

### 3.7. IRE1α Increases the Stability of MTDH Protein

Finally, we investigated if IRE1α contributes to the stabilization of MTDH. The previous experiments suggested that the stability of both IRE1α and MTDH is dependent on VCP activity. We compared CHX-treated parental HMC-1.2 with HMC-1.2^IRE1α−/−^ cells to investigate whether the presence of IRE1α affects the stability of MTDH. Indeed, the absence of IRE1α in the HMC-1.2^IRE1α−/−^ cells significantly reduced the stability of MTDH at an early time point (2 h) (Figure 6A,B). In conclusion, these data suggest an adaptive role of IRE1α in supporting the stability of the ER membrane protein MTDH. 

## 4. Discussion

MCL is known as the most aggressive form of SM. A key aspect of MCL, together with other types of cancer, is its enhanced translational activity and hence a susceptibility to an accumulation of misfolded proteins, resulting in constitutive activation of stress responses including the UPR. Suppressing the UPR has been identified as a potential strategy for cancer treatment [8,42]. Within this context, the role of IRE1 in modulating the balance between cellular survival and apoptosis under varying stress conditions is crucial but has not been fully elucidated for MCL. 

Our previous work demonstrated that pharmacological inhibition of IRE1 as well as its forced activation could attenuate proliferation and increase apoptotic cell death of the MCL cell line HMC-1.2. Here, we aimed to identify new interactors that are involved in the regulation of IRE1 function. To reach a higher sensitivity compared to immunoprecipitation approaches, we applied the BioID technique, which allows for the detection of weak and transient protein interactions. In particular, we made use of the advanced BioID variant TurboID [43], which has the advantage of requiring only a short (1 h) incubation period with biotin, to modify/identify bait (IRE1)-interacting proteins. However, the biotinylation of indirect interactors or proximal proteins that do not interact directly are usually present and need to be accounted for [44]. 

Although we mainly concentrated on the analysis of IRE1α-interacting proteins, we used this research work to identify IRE1β interactors as well and to define the differences between the IRE1α and IRE1β isoforms.

IRE1β has been reported to be uniquely expressed in epithelial cells lining mucosal surfaces [17]. There, it could negatively regulate IRE1α signaling in response to ER stress [18]. However, the presence of endogenous IRE1β protein was never confirmed due to the absence of specific antibodies. For the first time, we were able to detect the expression of *IRE1β* (*ERN2*) mRNA in human MCL cell lines (HMC-1.1 and HMC-1.2), and other cancer cell lines. IRE1β peptides were identified in a variety of MS data sets, including colorectal cancer tissues [45] as well as the NCI60 cell line panel [46]. Importantly, *IRE1β* gene deletion strongly decreased the basal *IRE1β* mRNA levels together with a reduction in *XBP1s* mRNA levels, suggesting the participation of IRE1β in *XBP1* splicing and target gene expression. 

The function of IRE1α depends on its dimerization/oligomerization and phosphorylation state. These are affected by different factors that tune the balance between IRE1α dimers and oligomers to determine the amplitude and kinetics of UPR signaling, for instance, the Sec61 core component of the translocon complex [47]. IRE1α forms a dynamic multi-protein platform termed the UPRosome [48] that can contact other protein complexes and organelles. Several cytosolic inhibitors, positive modulators, and post-translational modifiers of IRE1α have been identified. While Bax inhibitor-1 (TMBIM6) [49], fortilin [50], and BID [51] were identified as inhibitors, the BCL-2 family members BAX and BAK [52] can act as positive modulators. Moreover, it has been shown that the stability and activity of IRE1α can be modified by post-translational modifications like phosphorylation [53] or ADP-ribosylation [54]. 

Our analysis confirmed previously described interactors of IRE1α such as the chaperone HSP90 [55] and the translocon complex member Sec63 [47,56]. Moreover, several proteins involved in the ubiquitin-modification system were identified as IRE1-binding partners such as CHIP, ubiquitin D, USP14, and the UFM1-binding protein DDRGK1 [12,57]. 

We have additionally detected in our TurboID approach the E3 ligase UFL1 [58] as well as UBXN4, which connects IRE1α to VCP and ERAD [59]. IRE1α has been described as a target of the Sel1L-Hrd1 ERAD complex [41], which degrades IRE1α under basal conditions in a BIP-dependent manner.

Importantly, we have identified the AAA ATPase VCP as a direct interactor of IRE1α that affects IRE1α expression and degradation. VCP is a crucial component of the ubiquitin–proteasome system that plays a central role in ERAD. Its functions are to pull out misfolded proteins from the ER lumen and membrane for degradation, to extract ubiquitylated proteins from other membranes or cellular structures, and to segregate such proteins from interacting proteins to facilitate their degradation in the proteasome [60]. Interestingly, VCP is also involved in segregating transcription factors from chromatin and disassembling RNA–protein complexes [61]. The function of VCP is strongly dependent on its interaction with various cofactors, including ubiquitin adaptors, ubiquitin ligases, and deubiquitylating enzymes, to edit the ubiquitin chains on substrate proteins, thus determining their fate [60]. Active VCP is organized into a hexameric complex [39]; however, the technique used in our study does not allow us to differentiate between monomeric, dimeric, and hexameric VCP. Nevertheless, the effects of pharmacological VCP inhibitors point to the importance of active, hexameric VCP. 

Due to VCP’s central role in ERAD and the ability to induce irreversible proteotoxic stress through VCP inhibition, several VCP inhibitors are being tested in clinical trials [62]. The first-in-class VCP inhibitor CB-5083 has been demonstrated to exert cytotoxicity in vitro against canine lymphoma [63]. Interestingly, dogs commonly present with mast cell tumors in the skin [64], suggesting the efficacy of VCP inhibitors in such tumors as well. The survival and proliferation of cancer cells is strongly dependent on an adaptive UPR. This pro-survival mechanism can be switched to the pro-apoptotic, terminal UPR by forcing ER stress. This could be achieved by the inhibition of VCP, the main executer of ERAD, or the inhibition of the most conserved ER stress sensor IRE1α, which is mainly involved in protein homeostasis. Therefore, the combined inhibition of VCP and IRE1α could be more effective, allowing for the use of lower inhibitor concentrations and potentially attenuating side effects.

Mast cells release preformed mediators from so-called secretory lysosomes. We have identified the Vesicle-Associated Membrane Protein-Associated Protein A (VAPA), which plays a crucial role in vesicle trafficking, membrane fusion, protein complex assembly, and cell motility, as an IRE1 interactor. ER-resident VAPA forms ER–endosome contact sites via interactions with Oxysterol-Binding Protein 1 (OSBP) to mediate its lipid transfer function [65]. In addition, VAPA is also involved in autophagy [66] and ER-phagy, which are essential mechanisms in the UPR [67]. It has been shown in yeast that the soluble ER-phagy receptor Epr1 is upregulated via IRE1 signaling [68]. The selective autophagy receptor optineurin interacts with IRE1α, and optineurin deficiency amplifies IRE1α levels during ER stress [69]. Here, we described for the first time a direct interaction between VAPA and IRE1α in an MCL model system. Whether VAPA is involved in MC granule formation or autophagy needs further investigation. Interestingly, IRE1 also interacted with the nuclear membrane protein Emerin, suggesting that IRE1 also contacts the nuclear membrane. However, it has been reported that Emerin dissociates from the nuclear envelope during ER stress [70] and thus, IRE1 might interact with cytosolic emerin.

The interaction of MTDH with IRE1α and its destabilization by VCP offers an interesting view on tumor control by functionally interacting MTDH, IRE1α, and VCP. MTDH depletion or pharmacological inhibition disrupts the interaction with staphylococcal nuclease domain-containing 1 (SND1), which is required to sustain breast cancer progression in established tumors [15]. Pharmacological disruption of the MTDH–SND1 complex enhances tumor antigen presentation and synergizes with anti-PD-1 therapy in metastatic breast cancer [71]. Since we showed that VCP activity controls basal MTDH expression levels, the careful use of pharmacological VCP activators might attenuate the progression of breast cancer. Cytoplasmic MTDH has been shown to provide a survival advantage under conditions of stress by acting as an RNA-binding protein [15]. As mentioned before, SND1, which acts as a nuclease in the RNA-induced silencing complex (RISC) facilitating RNAi-mediated gene silencing, is an MTDH-interacting protein. Co-immunoprecipitation and co-localization studies confirmed that MTDH is also a component of the RISC, and both MTDH and SND1 are required for optimum RISC activity. Increased RISC activity, conferred by MTDH or SND1, resulted in increased degradation of tumor suppressor mRNAs that are targets of oncomiRs [72]. We have identified MTDH and SND1 as IRE1α interactors and showed that the stability of MTDH was increased by VCP inhibition, suggesting that MTDH is a target of the VCP-ERAD machinery. Its role as an RNA-binding protein and as a member of the RISC could have a potential impact on IRE1α-processed RNAs and needs further investigations.

An overview on the mentioned/analyzed factors and their potential functions on the basis of the relevant GO terms is shown in Appendix A.

## 5. Conclusions

IRE1α is a central regulator of the UPR, which is required, amongst other processes, for proteostasis in proliferating cancer cells. Using the technique of TurboID, we identified novel and verified already known IRE1α-interacting proteins in the KIT^D816V^-driven MCL cell line HMC-1.2. In particular, the interaction of IRE1α with VCP, a main constituent of the ERAD pathway, suggests a close functional interaction between the detection of misfolded proteins by IRE1α and the removal of such proteins from the ER by VCP. The development and use of pharmacological activators and inhibitors of IRE1α as well as VCP will allow for the careful titration of proteostasis and thus the optimization of the adaptive UPR or maximization of the terminal UPR for cancer therapy. Further identification and analyses of IRE1α-interacting proteins will enable the organ- and cell type-specific elimination of neoplastic cells.

## 6. Limitations of the Study

The study presented in this publication was mainly performed in the MCL cell line HMC-1.2, which expresses KIT^V560G,D816V^. Working with this cell line allowed us to conduct the analyses presented (TurboID-based identification of IRE1α interaction partners and verification by immunoprecipitation). The verification of the obtained data using primary leukemia cells from MCL patients would strengthen the conclusions of our study; however, due to limitations in obtaining sufficient cell numbers, this was not possible. Moreover, the biochemical verification of the differences/similarities between the IRE1α and IRE1β interaction partners identified in our BioID approach would have been preferable. Due to the unavailability of a suitable anti-IRE1β antibody, a confirmation of the endogenous levels was not possible. 

## Figures and Tables

**Figure 1 cells-13-00747-f001:**
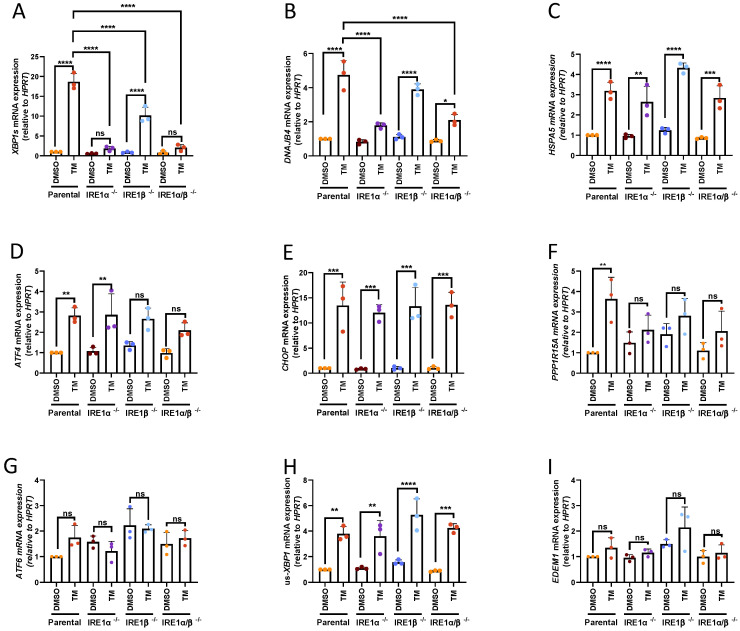
UPR activation in IRE1α- or IRE1β-deficient HMC-1.2 cell lines. Parental and IRE1-deficient (IRE1α−/−, IRE1β−/−, IRE1α/β−/−) HMC-1.2 cells were treated with vehicle (DMSO) or 5 µg/mL TM for 3 h. Deficiency in IRE1α protein in HMC-1.2^IRE1α−/−^ cells is shown in Figure 6. The expression of UPR target genes in the IRE1 pathway ((**A**) *XBP1s*, (**B**) *DNAJB4*, and (**C**) *HSPA5*), the PERK pathway ((**D**) *ATF4*, (**E**) *CHOP*, and (**F**) *PPP1R15A*), and the ATF6 pathway ((**G**) *ATF6*, (**H**) *usXBP1*, and (**I**) *EDEM1*) was evaluated by RT-qPCR and normalized to *HPRT* (*n =* 3). Data shown as mean ± SD. Significance levels are based on one-way ANOVA. * *p* < 0.05, ** *p* < 0.01, *** *p* < 0.001 **** *p*< 0.0001, ns not-significant.

**Figure 2 cells-13-00747-f002:**
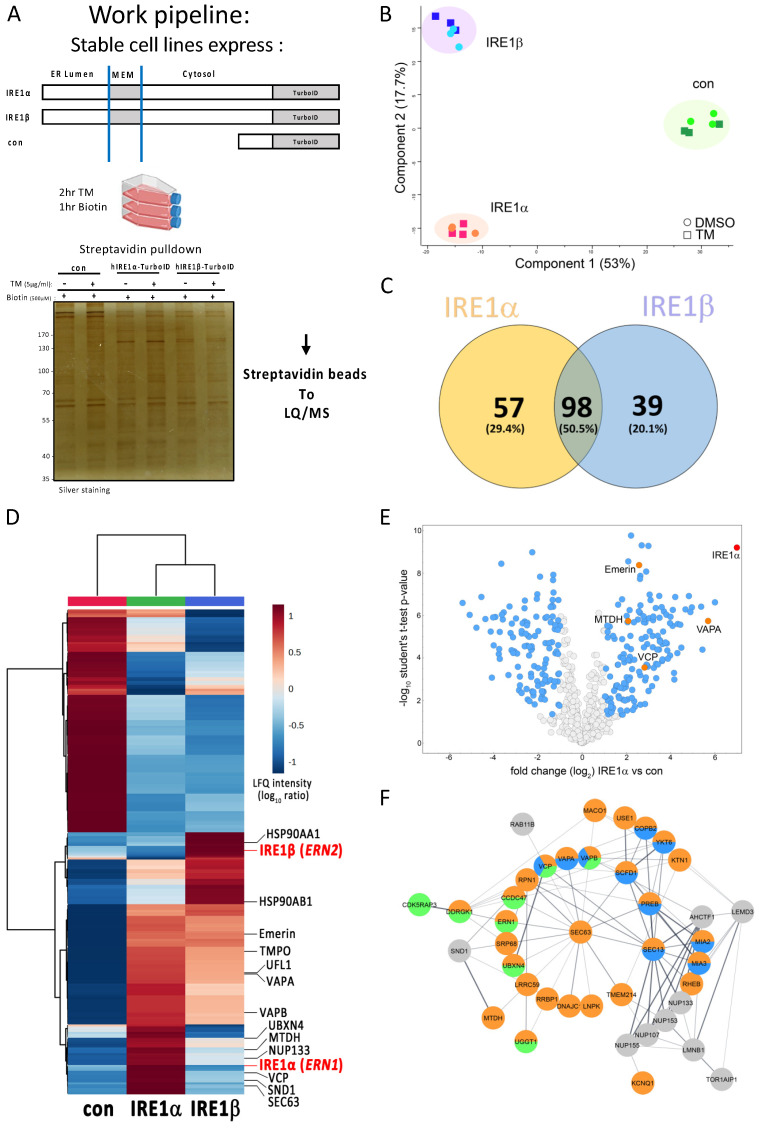
Identification of IRE1-interacting proteins with TurboID. (**A**) Generation of IRE1α-TurboID constructs and TurboID assay workflow. Schematic representation of IRE1 constructs and V5 control. TurboID was fused to the C-terminal domain of both human IRE1α (IRE1α-TurboID) and human IRE1β (IRE1β-TurboID). Parental HMC-1.2 and HMC-1.2^IRE1α−/−^ cells stably transduced with IRE1-TurboID target constructs were treated with TM (5 μg/mL) for 2 h before the addition of biotin (500 μM) for 1 h. Biotinylated proteins were collected on streptavidinmagnetic beads for the subsequent identification and quantification by mass spectrometry (LC-MS/MS). (**B**) PCA plot showing the segregation of the identified interactomes in DMSO- or TM-treated samples between the control (con, V5-TurboID) and the IRE1α/IRE1β cellular models across the two PCA axes. Data points of con (green symbols), IRE1α-TurboID (red symbols), and IRE1β-TurboID (blue symbols); three biological replicates are shown. Percentages correspond to the relative contribution of each axis to the overall variation in protein expression. (**C**) Venn diagram of 194 IRE1-interacting proteins shows unique and overlapping interactors of IRE1α- and IRE1β-TurboID fusion proteins in the presence of DMSO and TM. (**D**) Hierarchical clustering was performed with the top 250 overrepresented proteins in terms of change in significant LFQ intensity and enrichment compared to control, and the results are shown as a heatmap (distance is the Euclidean distance, and clustering algorithm using ward D). The fold changes were transformed to log_10_, and auto-scaling values were compared using one-way ANOVA with a *p*-value threshold of 0.01, with multiple comparisons corrected using statistical hypothesis testing (Tukey). (**E**) Volcano plot displays quantified protein interactors of IRE1α compared to the control. The x-axis displays log_2_ fold change of enriched proteins. Blue dots show proteins that were at least 2-fold enriched, orange dots show further investigated proteins, and the red dot indicates IRE1α. The y-axis displays log_10_ *p*-value with the threshold set at *p* < 0.05. (**F**) STRING analysis of the IREα interactors. All proteins were at least 2-fold upregulated in both IRE1α vs. control as well as IRE1α vs. IRE1β. The colors show the Gene Ontology analysis results of proteins with ER localization (GO:0005783; orange), involved in the ER stress response (GO:0034976; green) and ER to Golgi vesicle-mediated transport (GO:0006888; blue).

**Figure 3 cells-13-00747-f003:**
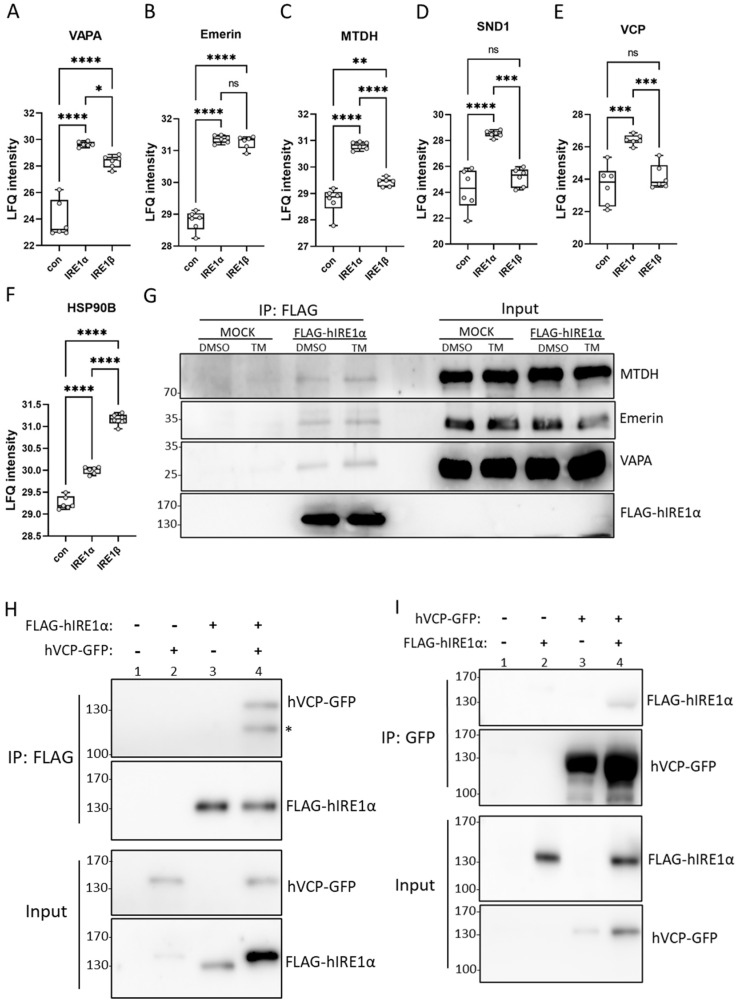
IRE1α interacts with proteins involved in membrane contact sites and protein stabilization. (**A**–**F**) Boxplots depict LFQ values of the interactors detected by LC/MS TurboID: VAPA, emerin, MTDH, SND1, VCP, and HSP90B, respectively. (**G**) HMC-1.2^IRE1α−/−^ cells stably transduced with FLAG-tagged human IRE1α or the empty vector control (MOCK) were treated with the solvent DMSO or TM (5 µg/mL) for 3 h. IRE1α was immunoprecipitated (IP) from cell lysates via an anti-FLAG antibody. Input and IP samples were immunoblotted for the detection of IRE1α, MTDH, emerin, and VAPA. (**H**,**I**) GFP-tagged VCP and FLAG-tagged IRE1α were overexpressed in HEK293 cells. IRE1α (α-FLAG) (**H**) and VCP (α-GFP) (**I**) were immunoprecipitated from lysates (Input) for further detection on immunoblots. (**H**) Top panel: notably, the band marked with an asterisk was not stably observed; bottom panel: this is the result of a consecutive detection, first with the α-GFP antibody and second with the α-FLAG antibody. hVCP-GFP is only slightly larger than FLAG-hIRE1α (approximately 2 kDa) and it was more strongly expressed. (**I**) As can be seen in the bottom panel, the expression of hVCP-GFP was inhomogeneous, which, consequently, was also observed in the second panel from the top upon immunoprecipitation. Data shown as mean ± SD. Significance levels are based on one-way ANOVA. * *p* < 0.05, ** *p* < 0.01, *** *p* < 0.001, **** *p* < 0.0001.

**Figure 4 cells-13-00747-f004:**
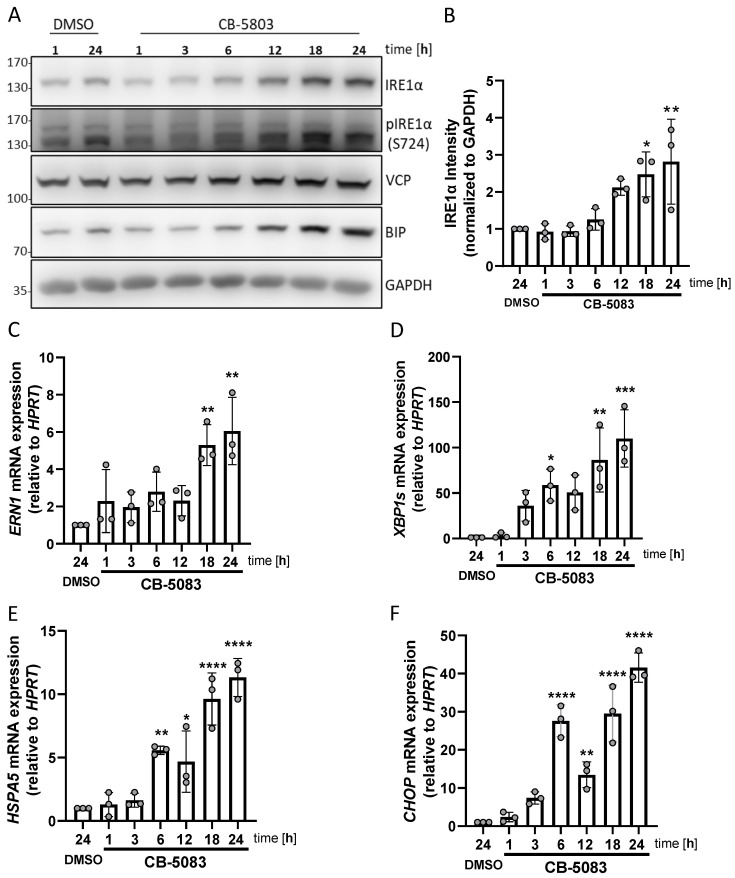
VCP inhibition stimulates UPR activation and IRE1α expression. (**A**) HMC-1.2 cells were treated with vehicle (DMSO) or 1 µM CB-5803 for the indicated time points. The phosphorylation of IRE1α (S724) and the expression of IRE1α, VCP, and BIP were detected by Western blotting. GAPDH served as the loading control. (**B**) Quantification (densitometry) of the data shown in (**A**) is the mean of three independent experiments. (**C**–**F**) HMC-1.2 cells were treated with vehicle (DMSO) or 1 µM CB-5803 for the indicated time points. *ERN1* (**C**), *XBP1s* (**D**), *HSPA5* (**E**), and *CHOP* (**F**) mRNA expression was evaluated by RT-qPCR and normalized to HPRT (n = 3). Data shown as mean ± SD. Significance levels are based on one-way ANOVA. * *p* < 0.05, ** *p* < 0.01, *** *p* < 0.001, **** *p* < 0.0001.

**Figure 5 cells-13-00747-f005:**
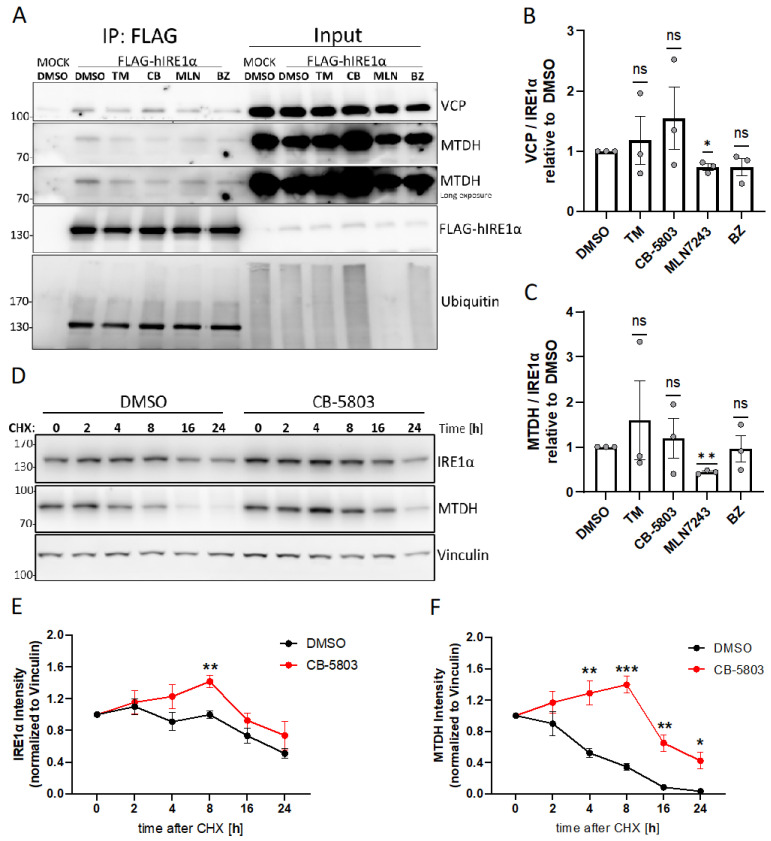
The interaction between IRE1α, VCP, and MTDH is dependent on ubiquitination. (**A**) HMC-1.2^IRE1α−/−^ cells stably transduced with FLAG-tagged human IRE1α or the empty vector control (MOCK) were treated as indicated with vehicle DMSO, 5 µg/mL TM, 1 µM CB-5803 (CB), 20 µM MLN7243 (MLN), or 10 nM bortezomib (BZ) for 3 h. Flag-tagged IRE1α was precipitated from lysates and analyzed together with the input on immunoblots for the detection of VCP, MTDH, IRE1α, and ubiquitin. Notably, the distinct bands seen on the left side of the ubiquitin panel are the IRE1α signal from the initial α-IRE1α detection prior to the incubation with the α-ubiquitin antibody. (**B**,**C**) Quantification (densitometry) of VCP (**B**) and MTDH (**C**) from blots in (**A**) is shown as the mean of three independent experiments. (**D**) Cycloheximide (CHX) chase assay in the presence of the VCP inhibitor CB-5803 (1 µM) or the solvent DMSO. CHX-treated HMC-1.2 cells were lysed at the indicated time points and IRE1α and MTDH were detected by Western blotting. Vinculin served as the loading control. (**E**,**F**) Quantification (densitometry) of three independent experiments. IRE1α (**E**) and MTDH (**F**) signals were normalized to the loading control and displayed values are relative to those in untreated cells. Error bars, s.e.m.; * *p* < 0.05, ** *p* < 0.01, *** *p* < 0.001, ns not-significant.

**Figure 6 cells-13-00747-f006:**
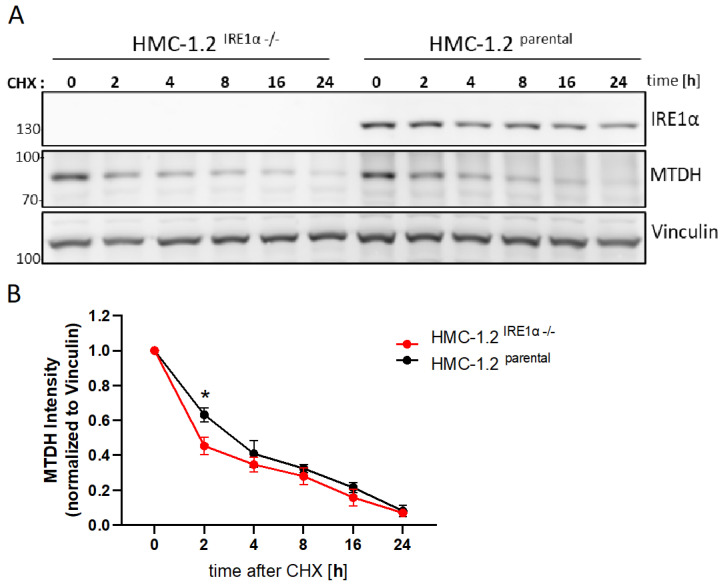
IRE1α increases the protein stability of MTDH. (**A**) CHX chase assays were conducted in parental HMC-1.2 and IRE1α-deficient cells. Cells were lysed at the indicated time points after CHX incubation and MTDH was detected by Western blotting. Vinculin served as the loading control. (**B**) Quantification (densitometry) of three independent experiments. MTDH signals were normalized to the loading control and displayed values are relative to those in untreated cells. Error bars, s.e.m.; * *p* < 0.05.

## Data Availability

The mass spectrometry proteomics data have been deposited to the ProteomeXchange Consortium via the PRIDE [73] partner repository with the dataset identifier PXD047343.

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
