# Peer review of "TurboID-Based IRE1 Interactome Reveals Participants of the Endoplasmic Reticulum-Associated Protein Degradation Machinery in the Human Mast Cell Leukemia Cell Line HMC-1.2"

_cells, 2024, doi:10.3390/cells13090747_

Round 1
Reviewer 1 Report
Comments and Suggestions for Authors
The MS addressed that pharmacological manipulation of IRE1α-MTDH-VCP complexes might be enable in the therapy of mast cell leukemia (MCL). The authors just used HMC-1.2 cells to perform these experiments. It found that the AAA ATPase VCP and the oncoprotein MTDH as IRE1α-interacting proteins. Pharmacological inhibition of VCP activity resulted in increased stability of the complex of IRE1α and MTDH. It is interesting and has novelty. But these results of MS are not enough to prove it. The MS need more experimental results and a major revision.
Major points:
1. In Fig.1, it need protein level of IRE1α and IRE1β in the KO cells, not only mRNA level.
2. In Fig.3, it need to detect the interaction of endogenous proteins in HMC cells. In Fig.3H, the bands of FLAG-h IRE1α in the Input Group have different size. Please check it. As well, the interactive domain of IRE1α need to identified for the interaction.
3. In Fig.5, the results is not enough to prove that the interaction between IRE1α, VCP and MTDH is dependent on ubiquitination (Ub). Fig.5A, it has no control to quantitative analysis. The Ub may be modified in the different proteins, including IRE1α, VCP and MTDH. It need to find which is important for the interaction.
4. In Fig.6, IRE1 increased the protein stability of MTDH just at Point 2h. It is hard to demonstrate the content. It need for further experiments.
5. The quality of WB image is not clear to show. Please check it.
6. The interaction of IRE1α-MTDH-VCP complexes in MCL cells need another HMC cells to perform these assays. It also need to set up the association between interaction and function phenotype.
Minor points:
1. The formal text of MS had no immunofluorescence (IF) data. But it had relative protocol.
2. In the study, it used lots of molecular inhibitors. It need to show the control effect of these inhibitors in HMC cells.
Comments on the Quality of English LanguageThe MS addressed that pharmacological manipulation of IRE1α-MTDH-VCP complexes might be enable in the therapy of mast cell leukemia (MCL). The authors just used HMC-1.2 cells to perform these experiments. It found that the AAA ATPase VCP and the oncoprotein MTDH as IRE1α-interacting proteins. Pharmacological inhibition of VCP activity resulted in increased stability of the complex of IRE1α and MTDH. It is interesting and has novelty. But these results of MS are not enough to prove it. The MS need more experimental results and a major revision.
Major points:
1. In Fig.1, it need protein level of IRE1α and IRE1β in the KO cells, not only mRNA level.
2. In Fig.3, it need to detect the interaction of endogenous proteins in HMC cells. In Fig.3H, the bands of FLAG-h IRE1α in the Input Group have different size. Please check it. As well, the interactive domain of IRE1α need to identified for the interaction.
3. In Fig.5, the results is not enough to prove that the interaction between IRE1α, VCP and MTDH is dependent on ubiquitination (Ub). Fig.5A, it has no control to quantitative analysis. The Ub may be modified in the different proteins, including IRE1α, VCP and MTDH. It need to find which is important for the interaction.
4. In Fig.6, IRE1 increased the protein stability of MTDH just at Point 2h. It is hard to demonstrate the content. It need for further experiments.
5. The quality of WB image is not clear to show. Please check it.
6. The interaction of IRE1α-MTDH-VCP complexes in MCL cells need another HMC cells to perform these assays. It also need to set up the association between interaction and function phenotype.
Minor points:
1. The formal text of MS had no immunofluorescence (IF) data. But it had relative protocol.
2. In the study, it used lots of molecular inhibitors. It need to show the control effect of these inhibitors in HMC cells.
Author Response
I have uploaded my report as a PDF.

Reviewer 2 Report
Comments and Suggestions for Authors
The study identified the relationship between the stability of IRE1α and the ER transmembrane protein Metadherin (MTDH) with Valosin-containing protein (VCP), a key component of the ERAD machinery. The presence of IRE1α was found to enhance the stability of MTDH, potentially playing a role in the progression of MCL (multiple myeloma).
The article can be improved in the following aspects:
1. The relationship and potential compensation between the three pathways of UPR (IRE1α, PERK, ATF6) were not discussed in the introduction. Overall, the introduction did not deeply explore the research background.
2. Figures S1C and S1D appear to be mislabeled and do not correspond to the descriptions in the original text. In Figure S1D, the mRNA levels of IRE1β appeared to increase after IRE1α knockout , but similar phenotypes were not observed in other results, and the authors did not provide an explanation for this result. Further clarification is needed.
3. Line 344: it was stated that "The expression of UPR target genes measured by RT-qPCR confirmed that TM was able to activate all three arms of the UPR (IRE1, PERK, ATF6) in parental HMC-1.2 cells." However, there was no corresponding data to support this conclusion.
4. The layout of Fig. 1 is disorganized and does not follow the order of appearance in the original text. The corresponding text is confusing and lacks a central idea.
5. In FigS2, the Flag tag was added to the N-terminus of IRE1. This is supposed to block the N-terminal signal peptide and lead to incorrect protein localization of the protein.
6. Fig2E only shows the volcano plot for IREα, and not for IREβ. The rationale for this selection should be provided, or the results for IREβ could be included in an additional figure.
7. In Fig3G, there were no bands observed for the Flag signal in the Input group. Additionally, in Fig3H, the Flag signal in the Input group exhibited noticeable fluctuations, but this phenomenon was not observed in the IP group. Furthermore, why was there a weak signal detected in the second lane of the WB where no Flag was present? These WB images need to be repeated or explained.
8. The description of Fig3 in the main text repeatedly used the word "specific", but the figure only shows the results for IRE1α. How can its specificity be demonstrated without conducting the same experiments for IREβ? Replicating the mass spectrometry results for IREβ is necessary to confirm its specificity.
9. The description of Fig4A in the main text states that the accumulation of phosphorylated signals is accompanied by the accumulation of IREα under drug treatment. However, the enhancement of phosphorylated signals in this case was simply due to increased total protein levels.
10. The protein accumulation shown in Fig4 was explained solely based on changes in mRNA levels, without considering protein degradation. However, the UPR system is closely related to degradation. Moreover, the focus on mRNA levels was not adequately explained.
11. The ubiquitin signal in Fig5A needs to be repeated and should not appear as a sharp band. Additionally, the assay should be performed under denaturing conditions to rule out the influence from co-precipitated proteins.
12. The conclusions drawn from Fig5A-C in the main text were not fully supported by experimental evidence that the ubiquitin signal was indeed weakened. The weakening of interaction appeared insignificant.
Author Response
I

Reviewer 3 Report
Comments and Suggestions for Authors
In the manuscript entitled “TurboID-based IRE1 interactome reveals participants of the 2 ERAD machinery in the human mast cell leukemia cell line 3 HMC-1.2”, the authors generated IRE1a and/or IRE1b knockout cells and conducted TurboID and mass spectrometry to identify IRE1 interacting proteins. They followed up with two of the new IRE1 interacting proteins, VCP and MTDH. They reported that VCP inhibition causes the increased stability of IRE1a and MTDH as well as IRE1a activation, whereas deletion of IRE1a destabilizes MTDH. The experiments are well designed and the results are in good quality. Several issues need to be addressed before the manuscript can be published.
Major points:
1. Some of the Western blot bands showed minor differences. For example, in Figure 5D, the control Vinculin bands showed variations. How are the bands quantified? What are the steps taken to ensure the accuracy in band quantification? Description of measurement of band density should be included.
2. Many factors have been described in the text, including known factors from literature and new ones identified by the TurboID. A figure that outlines the relations among these factors will be helpful, especially when they are discussed about their potential application in the cancer treatments. For example, in line 654 “a combination of VCP and IRE1a inhibitors might be even more effective”, a clear demonstration of the coordination of these factors will make the reading easier.
3. Figure 3G did not show any binding specificity for TM treatment. Why?
4. Figure 3H and 3I: The expression level of FLAG-hIRE1a is enhanced by hVCP-GFP. Why? The size of FLAG-hIRE1a is bigger with the coexpression of hVCP-GFP in Figure 3H but not in Figure 3I. Why?
Minor points:
1. Lines 631-632: “We have additionally identified in our TurboID approach the E3 ligase UFL1 [53] as well as UBXN4, which connects to VCP and promotes ERAD [54]” This sentence is confusing at the first glance. From the references, these have been reported before, so maybe “identified” is the wrong choice of word here.
2. Line 461: “E1 ligase inhibitor MLN7243”. Did you mean inhibitor of E1 ubiquitin activating enzyme?
3. Line 661: spell out OSBP
4. Line 90: change “is depending on” to “is dependent on”.
5. Lines 94-102 are irrelevant to the manuscript context.
6. S1C and S1D: the figures and legends do not match with each other, and the label of Fig in lines 333-339 do not match.
7. Line 585: add “the” before “enhanced translational activity.
Comments on the Quality of English LanguageSome editing is needed.
Author Response
I have uploaded my report as a PDF.
Round 2
Reviewer 1 Report
Comments and Suggestions for Authors
Many questions are answered simply by text, without the actually relevant exprimental data, as in fig 1, 5&6.
Comments on the Quality of English LanguageNo.
Author Response
Many questions are answered simply by text, without the actually relevant experimental data, as in fig 1, 5&6.
We would like to thank reviewer 1 for acknowledging that we could answer many of the raised questions by improving our explanations with respect to the data shown. In light of the time available for the revision process combined with the time required for experimental handling of the comments, it was not possible to respond to all the reviewers´ points with new experiments. Nevertheless, we were able to demonstrate IRE1α-VCP interaction in HMC-1.2 cells between endogenous proteins (shown as supplemental Figure 3D). Furthermore, we included a kinase-dead mutant of IRE1α (K599A) as well as a C-terminal deletion mutant (ΔIRE1α) lacking most of the potential ubiquitination sites, and showed that both mutants still were able to interact with VCP, suggesting that neither the kinase function of IRE1α nor the presence of the potential C-terminal ubiquitination sites within IRE1α are strictly necessary for this interaction (shown as supplemental Figure 3E). Moreover, we proved the functionality of the E1 ubiquitin activating enzyme inhibitor (MLN7243) (shown in supplemental Figure 5B).
In addition, we noted in the legend to Figure 1 that lack of IRE1α protein expression in IRE1α-deficient HMC-1.2 cells is demonstrated in Figure 6.
Moreover, to accommodate the lack of certain experiments, we have added a new chapter (6. Limitations of the study) to the revised version of our manuscript (marked in yellow). There, we focus on the problem of corroborating our data in cells from MCL patients, and the lack of IRE1β-specific antibodies.
Reviewer 2 Report
Comments and Suggestions for Authors
The authors largely answered my questions. Please make sure the information is also clear to readers.
Author Response
The authors largely answered my questions. Please make sure the information is also clear to readers.
We would like to thank reviewer 2 for her/his positive evaluation of the changes that we have conducted in the first round of revisions. To complete our manuscript and to guide the readers, we have added a new chapter (6. Limitations of the study) and marked it in yellow.